# Complexity of Cardiotocographic Signals as A Predictor of Labor

**DOI:** 10.3390/e22010104

**Published:** 2020-01-16

**Authors:** João Monteiro-Santos, Teresa Henriques, Inês Nunes, Célia Amorim-Costa, João Bernardes, Cristina Costa-Santos

**Affiliations:** 1Department of Community Medicine, Information and Health Decision Sciences—MEDCIDS, Faculty of Medicine, University of Porto, 4200-450 Porto, Portugal; teresasarhen@med.up.pt (T.H.); csantos@med.up.pt (C.C.-S.); 2Center for Health Technology and Services Research—CINTESIS, Faculty of Medicine, University of Porto, 4200-450 Porto, Portugal; imnunes@icbas.up.pt (I.N.); celcosta@med.up.pt (C.A.-C.); joaobern@med.up.pt (J.B.); 3Department of Obstetrics and Gynecology, Centro Materno-Infantil do Norte-Centro Hospitalar do Porto, 4200-450 Porto, Portugal; 4Instituto de Ciências Biomédicas Abel Salazar, University of Porto, 4200-450 Porto, Portugal; 5Department of Gynecology-Obstetrics and Pediatrics, Faculty of Medicine of University of Porto, 4200-450 Porto, Portugal; 6Centro Hospitalar Universitário de S. João, Alameda Hernâni Monteiro, 4200-101 Porto, Portugal

**Keywords:** labor, fetal heart rate, entropy, data compression, complexity analysis, nonlinear analysis, preterm

## Abstract

Prediction of labor is of extreme importance in obstetric care to allow for preventive measures, assuring that both baby and mother have the best possible care. In this work, the authors studied how important nonlinear parameters (entropy and compression) can be as labor predictors. Linear features retrieved from the SisPorto system for cardiotocogram analysis and nonlinear measures were used to predict labor in a dataset of 1072 antepartum tracings, at between 30 and 35 weeks of gestation. Two groups were defined: Group A—fetuses whose traces date was less than one or two weeks before labor, and Group B—fetuses whose traces date was at least one or two weeks before labor. Results suggest that, compared with linear features such as decelerations and variability indices, compression improves labor prediction both within one (C-Statistics of 0.728) and two weeks (C-Statistics of 0.704). Moreover, the correlation between compression and long-term variability was significantly different in groups A and B, denoting that compression and heart rate variability look at different information associated with whether the fetus is closer to or further from labor onset. Nonlinear measures, compression in particular, may be useful in improving labor prediction as a complement to other fetal heart rate features.

## 1. Introduction

Worldwide, approximately 15 million infants are born preterm (after less than 37 completed weeks of gestation) each year [1]. Over one-third of the world’s estimated 3 million annual neonatal deaths are related to preterm birth [2,3,4]. Even after surviving the neonatal period, infants born preterm are at increased risk of delayed childhood development and low economic productivity [5]. Therefore, interventions to reduce the preterm birth rate are of utmost importance.

Clinical decisions during labor and delivery in developed countries are strongly based on cardiotocography (CTG) [6,7,8], which has been one of the most used tools in assessing fetal wellbeing since the early ’60s. CTG combines fetal heart rate (FHR), obtained using a Doppler ultrasound probe or electrocardiogram electrodes, with uterine contractions (UC) measurements, obtained using an abdominal or intra-uterine pressure transducer. Both provide relevant information about the fetal condition and early detection of preterm labor and abnormal labor progress [7,9,10].

Despite the importance of assessing the wellbeing of the fetus and mother, poor agreement among physicians in the analysis and classification of CTGs is still a problem, even among experienced obstetricians, resulting in a high false positive rate [6,11,12]. In daily practice, FHR and UC are displayed on a printout or monitor to be visually interpreted by a clinician. Even when following specific, well-accepted guidelines (for example, the International Federation of Obstetrics and Gynecology (FIGO), associated with high sensitivity and low specificity [13]), interpretation of CTG relies on the clinician’s opinion and daily practice. This leads to a chance that adherence to conventional guidelines could be more harmful than beneficial [14].

The beat-to-beat variation of FHR reflects the influence of the fetus’ autonomic nervous system (ANS) and its components (sympathetic and parasympathetic) in the heart. Therefore, it is an indicator of the fetal pathophysiological status, which can be used in the assessment of fetal wellbeing [15] and its well-known influence on labor onset and progression [16]. A certain level of unpredictable fetal heart rate variability (fHRV) reflects sufficient capabilities of the organism in search of optimal behavior. Reduced fHRV is linked with limited capabilities and mental disorders [17]. The linear modeling approach is used to quantify sympathetic and parasympathetic control mechanisms and their balance through the measurement of spectral low- and high-frequency components. However, it has been shown that not all information carried by beat-to-beat variability can be explained by these components [18]. For this matter, in the past couple of decades, and with the fast development of computation, new signal processing and pattern recognition methodologies (namely entropy and compression) have been developed and applied to many different fields, including the analysis of fHRV [19,20]. These approaches can reveal relevant clinical information not exposed by temporal or frequency analysis [21].

Systems, such as Omniview SisPorto [22,23,24] and NST-Expert, which later became CAFE [25], can automatically deal with CTG assessment and then overcome the limitations of the visual assessment of CTGs mentioned above, but clinical judgment remains highly dependent on CTG analysis [26]. Since all FHR processing and analysis in these systems is based on morphological features provided by FIGO guidelines, they lack the integration of nonlinear indices that would allow them to be optimized.

The ability to predict preterm labor can improve the wellbeing of both fetus and mother. The successful prediction of preterm labor is an essential part of a decision support system for physicians to implement measures that adequately reduce related fetal morbidity and mortality (like the administration of corticosteroids to the mother in order to accelerate lung maturation and therefore decrease the risk of respiratory distress in the newborn).

The main objective of this work is to evaluate how useful nonlinear parameters, namely entropy and compression, can be as labor predictors by using antepartum FHR and UC traces one or two weeks before labor.

## 2. Materials and Methods

### 2.1. Nonlinear Methods

#### 2.1.1. Compression

The Kolmogorov Complexity (KC) [27] is defined as the function mapping a string x in an integer, bounded to a Turing Machine ϕ. The KC reflects the increase in new patterns along a given sequence. In this case, the word complexity refers to the algorithmic complexity, defined according to information theory, as the length of the shortest program p able to print the string x.
(1)$$KC_{\phi }\left(x\right)=\{\begin{array}{c} min\left\{\left|p\right|:\phi \left(p\right)=x\right\},\;if\;\phi \left(p\right)=x\\ \infty \;if\;p\;does\;not\;exit, \end{array}$$

For a random string, the output of the KC function will be the length of the original string, as any compression effort will end in information loss. On the other hand, the more reoccurring patterns, the less complex the string is.

Although this concept is objective, its applicability is limited by the fact that KC is not computable. Compressors are a close upper-bounded approximation of the KC function. For over 30 years, data compression software has been developed for data storage and transmission efficiency purposes. More recently, compression has been utilized in research fields like music, literature, internet traffic, and health [28,29,30].

In this work, we will assess the algorithmic complexity of FHR and UC signals by applying the Gzip compressor. Gzip [31] combines two classical algorithms—Lempel–Ziv (LZ77) [32], a dictionary based algorithm, and Huffman scheme [33]—by encoding sequences of high probability using shorter bits in comparison with lower probability strings, where longer bits are used. The amount of compression obtained depends on the input file size and the distribution of common substrings.

The idea is that for a given time series, the compression ratio (CR), i.e., the compressed size of the file divided by its original size, can be used to assess the complexity. A random series will have CR close to 1, whereas a series full of patterns will be highly compressible and, therefore, the CR will be close to 0. The Gzip with maximum compression levels and values presented represents the percentage of CR.

#### 2.1.2. Entropy

In 1991, Pincus developed the Approximate Entropy (ApEn), a regularity statistic tool used to quantify a system’s complexity based on the notion of entropy [34]. The ApEn measures the irregularity of time series and is defined as the logarithmic likelihood that the patterns of a time series that are close to each other will remain close when longer patterns are compared.

Later, in 2000, Richman and Moorman [35] proposed Sample Entropy (SampEn). Similar to ApEn, the SampEn measures time series irregularity. However, it does so with some major advantages: (1) self-matches are not counted, reducing bias; (2) it agrees much better than ApEn statistics with the theory for random numbers with known probabilistic character over a broad range of operating conditions; (3) the conditional probabilities are not estimated in a template manner. Instead, they are computed directly as the logarithm of conditional probability rather than from the ratio of the logarithmic sums, showing relative consistency in cases where ApEn does not [36].

To use either ApEn or SampEn, decisions on two different parameters, m, and r, have to be made. The m parameter is the embedding dimension, i.e., the length of sequences to be compared, while the tolerance parameter r works as a similarity threshold. Two patterns are considered similar if the difference between any pair of corresponding measurements is less than or equal to r. Values of 0.1, 0.15, or 0.2 standard deviations (SD) are usually used for parameter r, while *m* is mostly considered as 2 [37]. In this work, tolerance of 0.1 SD and an embedding dimension of 2 were used.

### 2.2. Data

The FHR data used for this study were from a retrospective cross-sectional study [38]. Each FHR trace corresponds to distinct fetuses from a singleton pregnancy. The selected traces were acquired between July 2005 and November 2010 during hospitalization in a tertiary care university hospital. All traces were acquired at least 48 h before delivery to guarantee they included no labor time. Furthermore, the traces included were at least 20 min long, during which the signal quality was over 80%, and the signal loss was less than 33%.

The cardiotocographic signals were acquired using an external ultrasound sensor applied to the maternal abdomen. The ultrasound signal is filtered, envelope rectified and digitized at a sampling rate of 800 Hz with a 12-bit precision [39]. Then, an autocorrelation function is used to calculate the heart period and the similarity between pulses of two consecutive heartbeats, as described in [40]. Via the digital outputs of the fetal monitors, resulting traces were analyzed using the Omniview SisPorto^®^ 3.7 system [23] at a sampling rate of 4 Hz (Figure 1).

SisPorto features used in this paper are summarily described in Table 1. Note that the SisPorto system does not perform any average or reduction in FHR/UC signals.

The 1072 traces selected ranged from 30 to 35 gestational weeks. Two groups were defined: Group A—fetuses whose traces date was less than two weeks before labor, and Group B—fetuses whose traces date was at least two weeks before labor. Physiological fetal and maternal features, such as maternal age (mAge) and baby gender, as well as some tracing characteristics such as trace duration and signal quality, were compared in both groups. Linear indices for uterine contraction analysis comprised of mean_UC (median of UC mean from 10min nonoverlapping blocks), sd_UC (median of UC standard deviation from 10min nonoverlapping blocks) and cv_UC (coefficient of variability of UC).

Two complexity measures, Gzip and SampEn, were considered in this work. Because the value of these measures depends on the trace size, each tracing was split into non-overlapping blocks of 10 min. Both Gzip and SampEn were computed for each block. Then, the median value of CR and SampEn for each fetus was used. Both complexity measures were calculated for FHR (Gzip_FHR and SampEn_FHR) and UC signals (Gzip_UC and SampEn_UC).

### 2.3. Statistical Analysis

Normality for continuous variables was evaluated by visual inspection of the frequency distribution (histogram). For normally distributed variables, the values for each group are presented as mean ± SD, and an independent samples t-test was performed. On the other hand, for skewed continuous variables, the values are presented as median (minimum-maximum), and the Mann–Whitney test was used to compare the two groups. The categorical variables were compared in the two groups applying the Chi-Square test or Fisher’s exact test as applicable.

Logistic regression, using Hosmer–Lemeshow to test the goodness of fit, was used to predict which fetuses will be born preterm in the next two weeks. Variables were selected using Wald’s backwards method. The concordance statistic (C-statistic), measured by the area under the receiver operating characteristic curve, was computed to assess the model’s discrimination.

Akaike Information Criterion (AIC), AIC=2k−2log(L), where *k* is the number of parameters and *L* the maximum value of the likelihood function, was used for model comparison, where a lower result suggests a better model.

Statistical analysis was performed with IBM SPSS Statistics for Windows, version 24 (IBM, Armonk, NY, USA).

## 3. Results

A total of 1072 antepartum tracings were used, 96 of which were born in the following two weeks (Group A). The main clinical characteristics of the group in which fetuses were born in the next two weeks (Group A) and the group in which they were not (Group B) are presented and compared in Table 2. Note that no differences were found between the groups for these variables.

SisPorto features were also compared between the two groups (Table 3). Statistical significance was found with variables iDec (*p* < 0.001), which was lower in fetuses who would be born in the next two weeks, and average long-term variability (abLTV), which was higher in fetuses who would be born in the next two weeks (*p* = 0.038).

Furthermore, while SampEn was not able to find differences between the traces from babies in the two groups with FHR and UC signals, Gzip was (*p* = 0.024 for FHR, *p* = 0.013 for UC), being lower in fetuses who would be born in the next two weeks (Group A) for FHR signals, while the opposite happened for UC signals. The standard deviation of UC was also significantly higher for Group A (*p* = 0.020).

Logistic regression, including all relevant variables (*p* < 0.05)—Gzip_FHR, Gzip_UC, sd_UC, iDec, a week of CTG (wCTG), and abLTV—was then performed using a backward selection model. The model obtained included the variables Gzip, iDec and a week of CTG (wCTG). Also, interactions between Gzip and wCTG were considered but found to be non-significant. Results from the logistic regression can be found in Table 4.

From this logistic regression model, abLTV and UC variables were removed from the initial set of predictors made by the model, and a C-statistic of 0.704 was obtained, with a 95% confidence interval range of 0.651–0.758. Also, the AIC obtained for this model was 603.763. The process was repeated considering all relevant physiological and linear features but without Gzip. This model, now without Gzip but with abLTV, achieved an AIC of 605.5 and a C-statistic of 0.691 (0.639–0.742).

The groups were also redefined and tested again. The same analysis as before was performed, except Group A consisted of fetuses who were born less than one week (instead of two weeks) from trace acquisition (n = 27, all preterm) and Group B consisted of all other fetuses (n = 1045, term and preterm babies), which were born as term and preterm babies. SisPorto and nonlinear features were compared between the groups, as carried out in our previous analysis (results in Appendix A).

The logistic regression results are shown in Table 5. Note that the same variables were included in the logistic regression.

This model achieved an AIC of 235.3 and a C-statistic of 0.728 (0.619–0.836), which is a small improvement compared with the first one described in this paper.

In Table 6, Spearman’s correlation coefficient between Gzip and different physiological measures of variability was calculated. Moreover, the same coefficient was calculated for each group. Statistically significant results were found for abLTV and avLTV for two weeks labor prediction.

## 4. Discussion

This study enhances the importance of the inclusion of nonlinear indices in clinical practice. In particular, the results suggest that the Gzip compression ratio, a measure of the time series complexity, may improve the predictability of labor onset when applied to FHR and UC signals.

The main objective of this work was to predict labor within two weeks. Both groups included preterm and term babies. In Group A, 46 of 90 were term babies, born between 36 and 37 weeks of gestational age; while in Group B, 44 of 976 fetuses were preterm. No statistical significance was found between term and preterm cases in Group A or Group B.

The information captured by compression relates to the information comprised of other physiological features, such as short and long term variabilities [41]. In our study, Gzip_FHR has a Spearman’s correlation coefficient of −0.524 and 0.5 with abSTV and avSTV’s variabilities, respectively. These results contrast with a previous study [41] where correlation values were much higher in absolute value (−0.851 and 0.774). Some different characteristics of the datasets used in each study can explain these differences. On the one hand, the dataset of our study was acquired in an antepartum setting, while the data from the previous study were recorded during the intrapartum. In line with this, the difference observed in the two studies suggests that compression looks at physiological regulatory mechanisms that differ between both settings. On the other hand, another possible explanation is the different sampling rates used in the two studies (4 Hz here, versus 2 Hz in the other study). This may indicate that some information is lost when using 2 Hz. This inkling is supported by the results of Gonçalves et al. [42], who found nonlinear differences between both sampling rates. However, the study of Gonçalves et al. [42] is an intrapartum study, and the tolerance parameter for entropy was computed using an automatic threshold proposed by Lu [43]. A multiscale analysis of scale two would be affected by the latter hypothesis (as it mimics a 2 Hz sampling rate), but in our study, no difference was found. Govindan et al. [44] suggested a different approach, modifying the definition of sample entropy using a time delay. Future studies should compare several methods to study the oversampling question.

When factoring by group, we found significant differences in correlations between Gzip and abLTV and avLTV (Table 6). Different studies [45,46,47,48] found HRV changes, such as variability increase and pattern formation throughout fetal maturation, captured by nonlinear indices. Here, different patterns arise in the two groups presented, meaning that compression attains different information from HRV when compared with usual metrics. However, no statistical significance was found in one-week labor prediction analysis. We believe this might be due to low statistical power, as the number of individuals in Group A was 27, making confidence intervals too wide.

Some papers [49,50] indicate different gender development throughout gestation and suggest taking this into account in model creation. Though it was taken into consideration, no significant results were found.

The mean compression ratio (instead of median) of the tracings’ block was also considered, and the results obtained were similar. These results suggest robustness of compression regarding skewness and outliers, as well as low intra-tracing variability. Furthermore, multiscale analysis [51] was also performed both for SampEn and Gzip up to five scales, since we were using intervals of 10 min (~1440 data points), but no improvement was found.

Two different definitions for the groups were tested. The same analysis as before was performed, considering Group A as babies who were born preterm less than two weeks, and then less than one week, from trace acquisition (n = 27). As shown in Table 4 and Table 5, the logistic regression included the same variables. A small improvement was verified when considering one week, compared with two weeks, from labor. These results reinforce the stability of compression when predicting labor time.

Nonlinear FHR features recognition is a problem in the clinical community because clinicians do not always know how to interpret it. Although entropy has been associated with the activity of central nervous system regulation [52,53], there are still no direct associations between compression and the fetus’ physiology. Compression looks for patterns in the series, and a healthy fetus is linked with a high compression ratio (a more chaotic signal leads to fewer patterns that are able to be compressed). In contrast, an unhealthy fetus, under the response of its regulatory system, creates a heart rate signal with more patterns, leading to a lower compression ratio. There is evidence that sympatho-vagal activity, and probably also central nervous system activity, are associated with the onset and progression of labor, namely via sympathetic activation and vagal inhibition mechanisms [16]. A continuous decrease in the sympathetic stress response during the last weeks before labor was also reported [54], contrary to a stable baseline sympathetic level. Being able to find links between these events and nonlinear indices is key for medical acceptance of these tools in daily practice. Therefore, it is imperative that a more thorough analysis of the FHR changes captured by compression is carried out in particular.

These results are relevant since an early prediction of labor as a decision support system for physicians can improve both fetus and mother assessment and care. In particular, being capable of predicting preterm labor is of extreme importance, as major risks to fetus and mother are associated with it.

This work has some limitations. The number of preterm cases is small, considering the week of the CTG variable is included. Because of this, only fetuses between weeks 30 and 35 of gestational age were selected, limiting the interpretability of the results. Although all the cases were hospitalized, no knowledge of the hospitalization cause is known.

Future studies should validate these models in larger datasets and, if possible, test them in different settings, such as during hospitalization and regular appointments.

## 5. Conclusions

Prediction of labor is of extreme importance since physicians will be able to take preventive measures to ensure that both baby and mother will be as prepared as possible. In this work, it was shown that nonlinear measures, compression in particular, can improve labor prediction.

## Figures and Tables

**Figure 1 entropy-22-00104-f001:**
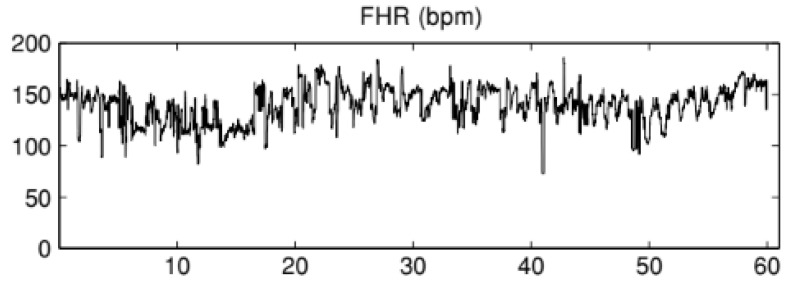
Example of a fetal heart rate (FHR) time series.

**Table 1 entropy-22-00104-t001:** Description of SisPorto features [22,24].

SisPorto Variable	Description
Basal line FHR	mean level of the most horizontal and less oscillatory FHR segments, in the absence of fetal movements and uterine contraction (UC), associated with periods of fetal rest, estimated via a complex algorithm
baseline	approximation of basal FHR to long-term FHR fluctuations using running averaging
number of accelerations (nAccel)	number of increases in FHR over the baseline lasting 15–120 s and reaching a peak of at least 15 bpm in 60 min
number of contractions (nContr)	number of periods in 60 min, lasting a maximum of 254 s, where an upward slope exceeding 17 s was detected reaching a peak lasting more than 90 s, followed by a downward slope exceeding 17 s
number of mild decelerations (mDec)	number of decreases in FHR under the baseline lasting 15–120 s, with a minimum amplitude of 15 bpm in 60 min
number of intermediate decelerations (iDec)	number of decreases in FHR under the baseline lasting 120–300 s, with a minimum amplitude of 15 bpm in 60 min
number of prolonged decelerations (pDec)	number of decelerations lasting more than 300 s in 60 min
average short-term variability (avSTV)	mean difference between adjacent FHR signals at 4 Hz on the fetal monitor, after removal of adjacent signals that differ >15 bpm
abnormal short-term variability (abSTV)	percentage of subsequent FHR signals differing <1 bpm
average long-term variability (avLTV)	mean difference between max and min FHR in a 1 min sliding window, in segments free of accelerations or deceleration
abnormal long-term variability (abLTV)	percentage of FHR signals with a difference between minimum and maximum values in a surrounding 1 min window <5 bpm

**Table 2 entropy-22-00104-t002:** Fetal and maternal features from Group A—fetuses whose traces date was less than two weeks before labor, and Group B—fetuses whose traces date was at least two weeks before labor.

	Group A (n = 96) Median (min-max), Mean ± SD or N (%)	Group B (n = 976) Median (min-max), Mean ± SD or N (%)	*p*-Value
Trace duration (min)	25.56 (14.82–67.07)	25.18 (11.28–96.31)	0.905
Gestational age at delivery (weeks)	36.58 ± 1.12	38.92 ± 1.20	
Maternal age (years)	31 (16–43)	31 (15–52)	0.291
Cesarean section	31 (32.3)	321 (32.9)	0.067
Baby presentation (cephalic)	90 (93.8)	918 (94.1)	0.524
Gender (male)	49 (51)	506 (51.8)	0.881
Signal quality (%)	97 (80–100)	96 (80–100)	0.105
Signal loss (%)	3 (0–20)	4 (0–21)	0.106

**Table 3 entropy-22-00104-t003:** SisPorto and nonlinear features from Group A—fetuses whose traces date were less than two weeks before labor, and Group B—fetuses whose traces date were at least two weeks before labor.

	Group A (n = 96) Median (min-max), Mean ± SD or N (%)	Group B (n = 976) Median (min-max), Mean ± SD or N (%)	*p*-Value
Basal line	133 (108–154)	134 (105–168)	0.137
Baseline	135.5 (114–160)	137 (105–169)	0.237
nAccel	5 (0–13)	5 (0–31)	0.188
nContr	1 (0–15)	1 (0–15)	0.200
mDec	0 (0–5)	0 (0–13)	0.787
iDec (% of no iDec)	89 (92.71)	962 (98.57)	**<0.001**
pDec (% of no pDec)	96 (100)	973 (99.69)	1.000
abSTV	50.49 ± 8.83	50.27 ± 8.42	0.805
avSTV	14.48 ± 3.48	14.55 ± 3.45	0.839
abLTV	1 (0–35)	0 (0–38)	**0.038**
avLTV	15.85 (8–33)	16.8 (0–40)	0.229
mean_UC	172.504 ± 103.426	166.663 ± 101.650	0.592
sd_UC	56.350 ± 42.403	45.768 ± 35.096	**0.020**
cv_UC	0.424 ± 0.347	0.369 ± 0.328	0.121
Gzip_UC	6.089 ± 1.769	5.664 ± 1.568	**0.013**
SampEn_UC	0.547 ± 0.306	0.595 ± 0.287	0.117
Gzip_FHR	11.559 ± 0.995	11.758 ± 0.878	**0.024**
SampEn_FHR	0.670 ± 0.159	0.693 ± 0.195	0.265

**Table 4 entropy-22-00104-t004:** Logistic regression for labor prediction in two weeks or less.

	B	*p*-Value	Exp(B)	95% CI
Constant	−20.639	<0.001		
wCTG	0.674	<0.001	1.962	1.489–2.584
Gzip_FHR	−0.341	0.005	0.711	0.560–0.902
iDec ^a^	1.782	<0.001	5.950	2.217–15.918

^a^ No iDec was set as reference instance.

**Table 5 entropy-22-00104-t005:** Logistic regression for labor prediction in one week or less.

	B	*p*-Value	Exp(B)	95% CI
Constant	−6.679	0.330		
wCTG	0.317	0.097	1.373	0.944–1.997
Gzip_FHR	−0.573	0.010	0.564	0.364–0.873
iDec	2.780	<0.001	16.112	5.205–49.874

**Table 6 entropy-22-00104-t006:** Spearman’s correlation coefficient and respective 95% confidence interval (CI) between Gzip_FHR and short- and long-term variabilities given by SisPorto. Confidence intervals were calculated using bootstrapping. Bold means significant differences between groups.

		Two Weeks Prediction		One Week Prediction	
	Total	Group A	Group B	Group A	Group B
abSTV	−0.524 (−0.564; −0.481)	−0.636 (−0.733; −0.501)	−0.512 (−0.565; −0.463)	−0.694 (−0.867; −0.370)	−0.515 (−0.560; −0.468)
avSTV	0.500 (0.452; 0.541)	0.596 (0.442; 0.720)	0.489 (0.437; 0.539)	0.698 (0.410; 0.864)	0.492 (0.444; 0.539)
abLTV	−0.562 (−0.602; −0.520)	−**0.722 (**−**0.807;** −**0.601)**	−**0.541 (**−**0.589;** −**0.495)**	−0.760 (−0.893; −0.489)	−0.551 (−0.596; −0.509)
avLTV	0.765 (0.737; 0.792)	**0.885 (0.818; 0.924)**	**0.751 (0.718; 0.780)**	0.874 (0.663; 0.970)	0.760 (0.730; 0.789)

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
