# Peer review of "Complexity of Cardiotocographic Signals as A Predictor of Labor"

_entropy, 2020, doi:10.3390/e22010104_

Round 1

Reviewer 1 Report

The main purpose of the study was to assess the usefulness of nonlinear measures of complexity (sample entropy and Gzip compression) of CTG signals as predictors of labor. The issues discussed are interesting and particularly important for improving of fetal diagnosis as well as for developing of automated systems to support physician decisions. The work is written carefully, with clearly defined objective, thorough experiments and conclusions. However, there are some questions that may need to be clarified. General and more specific comments are given below.

It was not indicated (or I overlooked it while reading) whether only one cardiotocographic signal was acquired for each fetus. It seems to me that it would be beneficial to compare the statistics of cardiotocographic parameters of such signals i.e., by analogically dividing them into two groups considering the week of the CTG variables. It seems that statistical tests were performed very carefully, so it is somehow surprising that there was no normality test for continuous variables (such as the Shapiro-Wilk test) and such an assessment was made only by visual inspection. It is desirable to present the results of statistical analysis (as in Tables 3 and 5) when evaluating groups of signals acquired in the period “less than” and “at least” one week before the labor. Studies comparing quantification of FHR variability using Doppler ultrasound and direct electrocardiography acquisition techniques undermine the ability to accurately determine FHR short-term variability using CTG. Therefore, in the absence of information on the accuracy of STV variability determination in the SiSPorto system, it is very difficult to validate the conclusions on the correlation between STV and Gzip_FHR. In my opinion, this is not a question of sampling frequency. The main problem for lack of common recognition of nonlinear FHR features by the medical community is their lack of interpretability. An attempt to outline how to understand the changes of the Gzip_FHR would significantly improve the practical value of the study.

In view of the above, I recommend accepting the paper after the comments have been addressed.

Reviewer 2 Report

In the manuscript titled “Complexity as predictor of labor” the authors propose the use of compression and entropy indices as labor predictors.  The topic seems of interest, however I have concerns regarding  the way in which the authors describe the  methodology and the results.  In my opinion, the actual methodology is hardly presented (what type of data are processed? how are they processed? why should the procedure work as a labor predictor?). Instead, the authors simply provide a superficial discussion of the notion of entropy and data compression. As for the latter, they refer to the “Gzip compressor”. This is a particular software implementation of the “deflate” algorithm which, in turn, is a combination of the classical Lempel-Ziv algorithm and the (also classical) Huffman coding scheme. For a research paper, a detailed description, with proper references, would be expected. In my opinion, the statistical analysis and the numerical results are not properly explained and displayed in the manuscript, either. Significant information is either missing or difficult to locate. 

All things taken together after reading the manuscript, it is not clear for me why these compression indices improve predictability and to what extent the do. Besides, the way the paper is currently written it may be a more suitable submission for a more specialised medical journal, rather than a larger-scope journal such as “Entropy”.

Round 2

Reviewer 2 Report

The authors have done some effort in clarifying the motivation of taking into account the indices of  entropy and data compression as predictors of labor and have included my previous suggestions and comments. In the revised version I suggest to label the axes of figure 1 with the magnitudes and their units, and complete the caption so that it provides a self-contained description of the figure.